# Microstructure and Wear Behavior of Ti-xFe-SiC In Situ Composite Ceramic Coatings on TC4 Substrate from Laser Cladding

**DOI:** 10.3390/ma17010100

**Published:** 2023-12-24

**Authors:** Xiaojun Zhao, Peize Lyu, Shenqin Fang, Shaohao Li, Xiaoxuan Tu, Penghe Ren, Dian Liu, Lyuming Chen, Lairong Xiao, Sainan Liu

**Affiliations:** 1School of Materials Science and Engineering, Central South University, Changsha 410083, China; zhaoxj@csu.edu.cn (X.Z.); 18890099737@163.com (P.L.); fangshenqin_yx@163.com (S.F.); lishaohao2022@163.com (S.L.); tuxiaoxuan@csu.edu.cn (X.T.); csuherb@csu.edu.cn (P.R.); liudian68@163.com (D.L.); clm2638372208@163.com (L.C.); 2Key Laboratory of Non-Ferrous Metal Materials Science and Engineering, Ministry of Education, Central South University, Changsha 410083, China; 3Center for Mineral Materials, School of Minerals Processing and Bioengineering, Central South University, Changsha 410083, China

**Keywords:** laser cladding, microstructure, TC4 substrate, coating, wear behavior

## Abstract

Titanium alloys are widely used in various structural materials due to their lightweight properties. However, the low wear resistance causes significant economic losses every year. Therefore, it is necessary to implement wear-resistant protection on the surface of titanium alloys. In this study, four types of in situ composite ceramic coatings with two-layer gradient structures were prepared on a Ti-6Al-4V (TC4) substrate using laser cladding. In order to reduce the dilution rate, a transition layer (Ti-40SiC (vol.%)) was first prepared on TC4 alloy. Then, a high-volume-fraction in situ composite ceramic working layer (Ti-xFe-80SiC (vol.%)) with different contents of Fe-based alloy powder (x = 0, 5, 10 and 15 vol.%) was prepared. The working surface of Ti-40SiC (TL) exhibited a typical XRD pattern of Ti, TiC, Ti_5_Si_3_, and Ti_3_SiC_2_. In comparison, both Ti-80SiC (WL-F0) and Ti-5Fe-80SiC (WL-F5) exhibited similar phase compositions to the TL coating, with no new phase identified in the coatings. However, the TiFeSi_2_ and SiC phases were presented in Ti-10Fe-80SiC (WL-F10) and Ti-15Fe-80SiC (WL-F15). It is proven that the addition of the Fe element could regulate the in situ reaction in the original Ti-Si-C ternary system to form the new phases with high hardness and good wear resistance. The hardness of the WL-F15 (1842.9 HV_1_) is five times higher than that of the matrix (350 HV_1_). Due to the existence of self-lubricating phases such as Ti_5_Si_3_ and Ti_3_SiC_2_, a lubricating film was presented in the WL-F0 and WL-F5 coatings, which could block the further damage of the friction pair and enhance the wear resistance. Furthermore, a wear-transition phenomenon was observed in the WL-F10 and WL-F15 coatings, which was similar to the friction behavior of structural ceramics. Under the load of 10 N and 20 N, the wear volume of WL-F15 coating is 5.2% and 63.7% of that in the substrate, and the depth of friction of WL-15 coating is only 14.4% and 80% of that in the substrate. The transition of wear volume and depth can be attributed to the wear mechanism changing from oxidation wear to adhesive wear.

## 1. Introduction

Due to their low density, high specific strength, high fatigue resistance, excellent corrosion resistance, and biocompatibility, titanium and titanium alloys are widely used in aerospace, vehicles, weapons, marine, biomedical, and petrochemical fields such as large aero engine parts, structural material for deep-water equipment, heat-exchange equipment, and medical alloy bone [1,2,3,4]. However, due to its low hardness and poor wear resistance, titanium alloys are becoming increasingly difficult to adapt to complex service conditions [5]. Therefore, it is necessary to solve the urgent problem of how to improve the surface hardness and wear resistance of titanium alloys and expand their range of application.

The main methods for the surface strengthening of titanium alloys include thermal spraying, cold spraying, plasma spraying, dip coating, electrophoretic deposition, and laser cladding [6,7,8,9]. Owing to the advantages of high energy density, good coherence, and excellent directionality, laser cladding as a surface-modification technology rapidly melts and solidifies the coating powder on the substrate surface through laser irradiation, thereby forming a fine and tough coating on the substrate surface [7]. Compared to other surface-strengthening technologies, laser cladding has several advantages: a good metallurgical bond with the substrate; a small dilution rate and heat-affected zone; fine microstructure due to large temperature gradients; and a material that is simple, flexible, and environmentally friendly [10].

Due to the extremely high hardness, high wear resistance, low density, and high designability, ceramic particles have often been added to metal to improve the mechanical strength and wear resistance of the laser cladding coating [11]. Ti-SiC systems are very prone to in situ reactions during the laser cladding process, thereby generating a TiC ceramic phase with high hardness and wear resistance, a Ti_3_SiC_2_ phase with self-lubrication, and a Ti_5_Si_3_ phase with good high-temperature resistance [12,13,14]. However, the addition of SiC particles is limited to less than 50%. To the authors’ knowledge, reports regarding the high-volume-fraction ceramics of laser cladding are rare in the literature. This is due to certain difficulties such as a high dilution rate and poor coating quality. Therefore, a transition layer and the different content of Fe modification are introduced into the coating in the present work. However, the questions remain of if and to what extent the high-volume SiC addition can improve the wear properties of laser cladding coating and what the wear mechanisms are. The aim of the present study was therefore to prepare the in situ composite gradient ceramic coating on Ti-6Al-4V(TC4) alloy using laser cladding to evaluate the microstructure, hardness, and wear properties and to identify the underlying frictional mechanisms.

## 2. Experimental Section

### 2.1. Raw Materials

A TC4 alloy sheet (Baoji Titanium Industry, Baoji, China) with a nominal composition of 90Ti-6Al-4V (wt.%) was used as the substrate in this work. The spherical Ti powder (40–110 μm, Beijing Ryubon New Material Technology Co., Ltd., Beijing, China), spherical SiC powder (60–180 μm, Shanghai Naiou Nano Technology Co., Ltd., Shanghai, China), and spherical Fe-based alloy powder (60–130 μm, Nangong City Yingtai Metal Material Co., Ltd., Nangong, China) with a purity of 4N were used as raw powders for laser cladding. The spherical Fe-based alloy powder had a nominal composition of Fe-13Ni-15Cr-3.2B-4.5Si-1.2C (wt.%). The TC4-based alloy in situ composite ceramic coatings were prepared using the coaxial powder transfer laser melting process shown in Figure 1.

### 2.2. Preparation of the Coating

The TC4 alloy substrate with dimensions of 50 mm × 50 mm × 6 mm was cut using the Electrical Discharge Machining (EDM) wire-cutting technique (DK-77, Kaiguang Precision Machinery Co., Ltd., Suzhou, China). In order to remove the oxidation on the surface, the substrate was ground with sandpapers and ultrasonically cleaned with an alcohol solution for 5 min. The thoroughly mixed powders in a volume percentage, as shown in Table 1, were attrition-milled for 12 h using ethanol as a milling medium, with a milling rate of 120 rpm.

The coatings were prepared through laser cladding in an argon atmosphere (oxygen content of less than 500 ppm) equipped with a coaxial powder feeding system and six-axis robotic system, using a laser focus spot diameter of 3 mm, a power of 1200 W, a laser scanning speed of 8 mm/s, a power disk speed of 1 r/min, and a lap rate of 30–40%. Finally, the prepared coating with a TC4 substrate was air-cooled down to room temperature.

### 2.3. Microstructure Characterization

The phase and crystallinity were analyzed via X-ray diffraction (XRD, Rigaku D/Max 2500, Rigaku, Tokyo, Japan) using Cukα1 radiation at 45 kV and 40 mA (λ = 1.540 Å). The diffraction angle (2θ) at which the X-rays hit the specimen varied from 20 to 80° with a scanning speed of 8°/min. The corrosion of the specimens was carried out in a mixture of HF:HNO_3_:H_2_O = 1:2:7 for 5 s. The microstructures were examined using a field emission scanning electron microscope (FESEM, MIRA3 LMH, TESCAN Co., Brno, Czechia). The elemental concentration profile of the microstructure was determined using energy-dispersive X-ray spectroscopy (EDS One Max 20). The Vickers hardness was determined on a polished cross section using a Vickers hardness tester (200HBVS-30, Changsha Huayin Testing Instruments Co., Ltd., Changsha, China) at a load of 10 N for a dwell time of 15 s. Twenty indentations were taken in each 100 μm on each sample and the average values were reported. In addition, the hardness distribution of the cross-sectional specimen was observed using an optical microscope (OM, Leica Microsystems, Wetzlar, Germany).

### 2.4. Frictional Wear Experiment

The samples were cut into specimens 20 × 30 × 6 mm^3^ in size using an EDM wire-cutting technology and polished with 80-mesh, 150-mesh, 240-mesh, 400-mesh, 600-mesh, 800-mesh, and 1000-mesh diamond grinding discs in order to make the surfaces smooth and flat. And the values of surface roughness (Ra) around the wear mark of each sample were all below 1.6 μm. The frictional wear experiments of the samples with a dimension of 20 mm × 30 mm × 6 mm were performed using the ball-on-disk tribometer (HT-1000 tester, Zhongkekaihua science technology Co., Ltd. Lanzhou, China) at room temperature (20 ± 2 °C), with Si_3_N_4_ balls with a diameter of 6 mm on the pins, normal loads of 10 N and 20 N, rotational speeds of 300 r/min, and a time of 30 min, and the experiment was repeated three times to reduce errors. The coefficients of friction (CoF) were obtained automatically using the device during the experiments. The wear surface and wear debris were observed using FESEM of MIRA3 TESCAN type equipped with an EDS spectrum analyzer, and the relevant areas were analyzed for elements. The wear surfaces were examined using an optical profilometer (Contour GT-K, Bruker Nano, Billerica, MA, USA), and three-dimensional profile measurements of the wear tracks and friction curves were plotted.

## 3. Results

### 3.1. Microstructure

#### 3.1.1. Microstructure of the Transition Layer

The XRD patterns of the TL coating are shown in Figure 2. It can be seen that the obtained coating exhibited a typical XRD pattern of four primary phases, including Ti, TiC, Ti_5_Si_3_, and Ti_3_SiC_2_, which is consistent with the results of the Ti-Si-C ternary phase diagram [15]. Furthermore, it is of interest to observe that no SiC was present after laser cladding, indicating that the addition of SiC was thoroughly reacted and transformed to TiC, Ti_5_Si_3_, and Ti_3_SiC_2_.

The cross-sectional microstructure of the TL coating is shown in Figure 3, and the corresponding EDS results are tabulated in Table 2. As shown in Figure 3a, there was a distinct interface between the substrate and the coating without surrounding defects, indicating a good metallurgical bond between them. The genesis of holes at the top region of the TL coating could be ascribed to the existence of gas-containing bubbles within the ceramic powder. During the cooling process of the laser melting, the liberation of gases led to the formation of holes. The microstructure in the bottom of the coating is shown in Figure 3b, consisting of a dendritic, inter-dendritic microstructure. According to the EDS results, the atomic ratio between Ti and C of point A is 1:1, confirming the dendrites (point A) are in the TiC phase. Compared to the direct addition of ceramic phase particles in laser cladding [16], the in situ TiC-reinforced phase showed a distinctive dendritic structure at the bottom of the coating. This is due to the fact that TiC is susceptible to dendritic growth under the action of a solid–liquid interface instability [17,18]. The inter-dendritic organization (point B) contains a notably higher content of Si than TC4 substrate. This can be attributed to the enrichment of Si in the liquid phase prior to the formation of the solid–liquid interface, leading to the Ti_5_Si_3_ phase being precipitated along the grain boundaries and the formation of a network of the inter-dendritic organization [19].

It can be seen in Figure 3c that the middle region of the coating consisted of granular, dendritic, and inter-dendritic microstructure. According to the EDS results, the granular phase (point C) is TiC. Within the middle region, the granular phase began to exist in the form of spherical aggregation. This suggests that the spherical SiC particles would react with the Ti element before the onset of the Marangoni effect. As shown in Figure 3d, the top region of the coating consisted of a coarser dendritic, inter-dendritic, and granular microstructure. According to the EDS results in Table 2, the atomic ratio between Ti and Si of the coarser dendrites (point D) is close to 5:3, indicating that the generated phase is Ti_5_Si_3_. The increase in inter-dendritic organization leads to the formation of dendrite Ti_5_Si_3_ structures, which is consistent with the Ti_5_Si_3_ phase in the laser cladding of the Ti-Si system [20].

Additionally, it should be pointed out that no conspicuous Ti_3_SiC_2_ phase is observed in Figure 3, although it is detected in the XRD patterns, as shown in Figure 2. This might be attributed to the relatively low concentration of the Ti_3_SiC_2_ phase.

#### 3.1.2. Microstructure of the Working Layer

The XRD results of the four prepared types of Ti-SiC coatings with different Fe-based alloy additions are shown in Figure 4. It can be seen that with the increase in Fe-based alloy addition, the phase content and phase composition were changed accordingly. When the added Fe-based alloy was less than 5%, the phases were similar to the TL coating. This means that a limited quantity of Fe-based alloy added into the coatings did not result in the emergence of novel phases, which suggests that a part of the Fe, Cr, and Ni elements might be solidly dissolved in Ti [21,22]. However, it could be seen that the intensity of the Ti peak was significantly decreased, while the intensity of the ceramic phase peaks was increased. It is noteworthy that the presence of SiC was still not detected in either the WL-F0 or the WL-F5 coating, indicating that all the SiC particles entering the melt pool had completely reacted.

When the addition of Fe-based alloy was increased to 10%, the coating exhibited the typical XRD patterns of TiC, SiC, and TiFeSi_2_ phases. The original silicide ceramic phases, such as Ti_5_Si_3_ and Ti_3_SiC_2_, disappeared in the XRD diffraction pattern, replaced by the TiFeSi_2_ phase. This implies that the addition of a sufficient amount of Fe-based alloy induced new phase reactions in the laser cladding process. When the added Fe-based alloy was further increased to 15%, the phases were similar to the WL-F10 coating. However, the intensity of the TiC phase peaks was increased, which indicated that the carbide ceramic phases with superior mechanical properties exhibited a higher content in the WL-F15 coating.

The cross-sectional microstructure of the four coatings is shown in Figure 5, and the corresponding EDS results are tabulated in Table 3. It can be seen that the thicknesses of the four coatings were all above 1000 μm, which was significantly thicker than that of the TL coating. After the preparation of the working layer, there was a significant reduction in the thickness of the transition layer, indicating the working layer had a high dilution rate (40–60%). If the working layer is prepared directly on the substrate, its dilution rate will be higher, and an excessively high dilution rate will reduce the performance of the substrate and cause serious thermal deformation. Moreover, in the outermost regions of the WL-F10 and WL-F15 coatings, an adhesion layer became apparent. Cracks, holes, and bubbles could be clearly seen in the adhesion layer (as shown in the lower left corner of Figure 5c,d), which signified its poor quality. Furthermore, the adhesion layer contained fewer gray ceramic phases compared to the working layer. Instead, it primarily consisted of a substantial quantity of SiC fragments (point H) and unreacted SiC spherical particles, which indicated that SiC reacted incompletely within this layer. The primary reason for the formation of this layer is that the Fe-based alloy has a more extended melting time range and higher SiC wettability than Ti, which could facilitate the SiC particles entering the melt pool and increase the thickness of the coating. Due to the rapid cooling at the surface of the melt pool during the laser cladding process, some of the entering SiC particles did not undergo complete reactions. Also, some defects such as holes and agglomerates were in the cladding portion of the coating, which could be attributed to the adverse consequence of the excessive addition of ceramic powder. For ceramic/metal composites with a high-volume-fraction of ceramics, the ceramic/metal interfaces may become weak connections, leading to a significant reduction in the interface bond, which leads to the formation of defects [23]. However, with the increase in Fe-based alloy addition, the defect content in the working layer decreased. This can be attributed to the self-fluxing nature of Fe-based alloy powders, which effectively mitigates the formation of holes and facilitates crack healing [24].

As shown in Figure 5(a1,b1), the top regions of the WL-F0 and WL-F5 coatings consisted of a strip, coarser dendritic, dendritic, granular, and inter-dendritic microstructure. According to the EDS results shown in Table 3, the Si content of the strip phase (point A) is significantly lower than Ti_5_Si_3_, and the atomic ratio of Ti:Si:C is close to 3:1:2, which is speculated to be Ti_3_SiC_2_. Ghosh et al. [25] studied an investigation of Ti_3_SiC_2_ through an in situ spark plasma sintering process, where the phase structure of Ti_3_SiC_2_ also exhibited strip- and plate-shaped morphology. The dendrites (Point B) are TiC, and the coarser dendrites (Point C) are Ti_5_Si_3_. The addition of Fe could provide more liquid phase for the formation of Ti_5_Si_3_ and effectively reduce the size of the Ti_5_Si_3_ precipitate phase [26], facilitating the precipitation of the Ti_5_Si_3_ phase. This indicates the reason for the higher content of inter-dendritic organization in the WL-F0 coating compared to the WL-F5 coating, while the ceramic phase content is significantly lower than that in the WL-F5 coating.

As shown in Figure 5(c1,d1), microstructures were completely different from those of WL-F0 and WL-F5 coatings. The enlarged area of the WL-F10 coating (Figure 5(b2)) revealed three phases: white matrix phase (point D), gray phase (point E), and black fine granular phase (point F). According to the EDS results shown in Table 3, the atomic ratio between the Ti and C of the gray phase is 1:1, indicating that the formed phase is TiC; the white matrix phase contains Ti, Fe, Si, and C, and based on the results of the atomic ratio in the EDS and the results of the XRD diffraction, it can be inferred that the composition of the white matrix phase is TiFeSi_2_; the black fine granular phase was generated at the boundary between the TiFeSi_2_ and TiC phases, which can be ascribed to be the SiC phase. The enlarged area of the WL-F15 coating (Figure 5(c2)) showed a “flower-like” dendritic (point G), white-phase, and black fine granular microstructure. According to the EDS results, this “flower-like” dendritic phase is TiC. Emamian et al. [27] studied an Fe-TiC coating prepared using laser cladding, and a similar TiC organization of dendrites was found in the cladding zone.

In order to gain a comprehensive understanding of the elemental distribution of Ti, Si, C, Fe, and Al elements within the coatings, compositional surface scans were performed on the enlarged regions (indicated in Figure 5(a2–d2)), as shown in Figure 6. In the WL-F0 and WL-F5 coatings, Ti elements exhibited a widespread distribution throughout the coating, without significant aggregation. Si elements were aggregated in coarser dendrites (Ti_5_Si_3_) and strip phases (Ti_3_SiC_2_), which could be clearly seen in the dotted line area of Figure 6a,b. With the increase in addition of the Fe-based alloy to 10%, a significant reduction in the Ti elements content could be observed in the area of the dotted line (Figure 6c), and this area showed an enrichment of Fe and Si elements. C elements were mainly aggregated within the gray granular phase (TiC) and the black fine granular phase (SiC). With the increase in Fe-based alloy addition to 15%, Ti and C elements exhibited aggregation and overlapped within the flower-like dendritic structure (dotted line area of Figure 6d), whereas Fe and Si elements were primarily distributed within the matrix phase. Al, as a self-contained element in the substrate, had low content in the coating, which only serves as collateral evidence for the distribution of Ti elements.

With the increases in Fe-based alloy content, significant changes occurred in the elemental distribution within the coating. Most notably, Ti elements transitioned from widespread distribution to aggregation, while Si elements transitioned from aggregation to widespread distribution. This transformation can be attributed to the conversion of Ti-rich silicide ceramic phases (Ti_5_Si_3_, Ti_3_SiC_2_) to the Ti-poor matrix phase (TiFeSi_2_), which indirectly leads to an increase in the content of hard ceramic TiC phases.

### 3.2. Properties

#### 3.2.1. Vickers Hardness

Figure 7 presents the change in the Vickers hardness of the cross-sectional coatings. Compared to the substrate, each group of coatings exhibited varying degrees of improvement in hardness. As shown in Figure 7a,b, the hardness of the working layer in the WL-F0 and WL-F5 coatings ranged from 749.2 to 622.2 HV_1_ and 1031.1 to 686.2 HV_1_. The hardness of the WL-F5 coating was shown to be slightly higher than that of the WL-F0 coating, which could be attributed to the increased ceramic phases in the working layer. As shown in Figure 7c,d, with the further increase in Fe content, both the maximum hardness of the working layer in the WL-F10 and WL-F15 coatings showed a significant increase, reaching 1420.7 HV_1_ and 1842.9 HV_1_, which was 4.06 and 5.26 times greater than that of the substrate and 1.90 and 2.46 times greater than that of the WL-F0 coating. Additionally, adhesion layers were present in the outermost of the WL-F10 and WL-F15 coatings, which showed the characteristics of low hardness and susceptibility to cracking. This is due to the poor bonding of the adhesion layer, which should be removed during subsequent frictional-wear experiments. Additionally, the decline in hardness within the working layer was severe, gradually slowing down in the transition layer, which indicated that the decreasing trend of the concentration of the hard phase in the working layer along the substrate direction is much larger than that in the transition layer.

The hardness results showed a significant rise with the increase in Fe-based alloy content. This illustrates that the granular or flower-like dendritic phase TiC enveloped by the TiFeSi_2_ matrix phase has higher hardness compared to traditional Ti-Si-C system and Fe-Ti-C system [27] laser cladding coating.

#### 3.2.2. Frictional Wear Property

Figure 8 shows the CoF curves and the average CoF values in frictional-wear experiments under loads of 10 N and 20 N. According to the CoF curves, the sliding friction process can be divided into two stages, including the initial wear stage and stable wear stage [28]. The CoF of TC4 substrate under the load of 10 N experienced gradual increments during the initial wear stage and stabilized in the subsequent stable wear stage. As the Fe content ranged from 0 to 5%, the average CoF values were significantly lower than that of the TC4 substrate, indicating that the coatings have a certain wear-reducing performance. With a further increase in Fe content from 10 to 15%, the CoF values showed a longer initial friction stage and were slightly higher than those of WL-F0 and WL-F5 coatings, while both of them were still lower than that of the TC4 substrate.

Under a 20 N load, the average CoF of the TC4 substrate remained basically unchanged, but the fluctuation increased compared with that under a 10 N load. The CoF values of the four coatings were suddenly elevated in a short initial wear stage and were similar and stable, and significantly lower than that of the TC4 substrate. This phenomenon indicates that each group of coatings had comparable wear-reducing performance under the 20 N load.

Figure 9 shows the SEM images and 3D morphologies of the wear surfaces of substrate and coatings at a 10 N load. As shown in Figure 9(a,a1,a2), the wear surfaces of the TC4 substrate had obvious destructive phenomena, such as plastic deformation, tear marks, and a typical large number of long and deep continuous furrows, which are the typical features of adhesive and abrasive wear [29,30]. Micro-plowing could be found on the wear surface [31], and the width and depth of micro-plowing were much smaller than those of furrows, which might be caused by fine wear debris (as shown in the lower left corner of Figure 9a). The poor wear resistance of the TC4 substrate is due to its low surface oxide protection, low hardness, low resistance to plastic shear, and susceptibility to adhesion transfer [32].

In contrast to the TC4 substrate, the wear surfaces of the WL-F0 and WL-F5 coatings exhibited scar regions, as shown in Figure 9b,c. This is due to the friction-generated heat accumulation at the friction pair/coating contact interface, resulting in a rapid increase in temperature and mutual adhesion, with the joints being pulled down to form scars during the subsequent sliding process [33]. Figure 9(b1,c1) shows the magnified SEM images of the scar regions of the WL-F0 and WL-F5 coatings. There is an intergranular fracture phenomenon around the scar area in Figure 9(b1), which is caused by the fracture of the brittle ceramic phase [34]. By observing the 3D morphologies (Figure 9(b2,c2)), it can be found that the wear surfaces of two coatings were smoother and the depths of friction were shallower compared to the TC4 substrate. It is hypothesized that silicide ceramics, such as Ti_5_Si_3_ and Ti_3_SiC_2_, have good self-lubricating properties, which enable them to form a lubricating film during friction to achieve wear resistance [13]. The wear mechanisms of the WL-F0 and WL-F5 coatings are abrasive and adhesive wear.

As shown in Figure 9d,e, the wear surfaces of the WL-F10 and WL-F15 coatings had obvious oxidation films. The delamination part of the oxidation film possessed the characteristic of non-conductivity, which was a very obvious of oxidation wear. Obvious cracks in the oxidation film area of the WL-F10 coating (Figure 9(d1)) can be observed, which were caused by the non-dense oxidation film under load. The delamination area of the WL-F15 coating (Figure 9(e1)) notably increased in comparison to the WL-F10 coating. According to the EDS results shown in Table 4, the delamination areas (points A and B) of the oxidation film were SiO_2_ + TiO_2_. Observing the 3D morphologies (Figure 9(d2,e2)) shows that the WL-F10 and WL-F15 coatings showed reduced wear volume loss, and the underlying wear surface morphology appeared slightly roughened, which can be attributed to the formation of oxidation film during the wear process. This oxidation film, as an interference layer, can effectively prevent the friction pair from causing deeper friction damage to the coating, greatly enhancing wear resistance [35,36]. The wear mechanism of the coatings is oxidation wear. The wear debris SEM images of the four coatings are shown in the lower-left corner of Figure 9. In comparison to the substrate, these coatings generated a smaller volume of lamellar wear debris and less fine wear debris.

Figure 10a shows the wear-track section curves of coatings and substrate at a load of 10 N, and Figure 10b shows the wear volume and depth of friction of coatings and substrates at a load of 10 N. The wear volume (*V*) in a frictional wear process can be calculated according to the following equation [31]:(1)V=LA
where *A* is the integral of the wear track section curves and *L* is the frictional wear distance.

The depth of friction, width of friction, and wear volume of the four groups of coatings exhibited significant reductions compared to the substrate. The decrease in WL-F0 and WL-F5 coatings indicated that the lubrication film formed by the self-lubricating phase played a certain wear-resistant role while avoiding the generation of furrows. An appropriate amount of dendritic structure can effectively improve wear resistance, but excessive dendritic structure may lead to a decrease in wear resistance [37]. This is because a certain content of dendrites can serve as a supporting surface to improve wear resistance [38], but they are prone to fragmentation when the dendrite content is excessive [39]. An excessive dendritic ceramic phase brings high hardness and shallow wear depth to the WL-F5 coating but also increases wear width and curvature radius due to the fragmentation and peeling of dendrites, ultimately leading to an increase in wear volume. The WL-F10 and WL-F15 coatings exhibited superior wear performance due to the protection of the oxidation film. The WL-F15 coating had the best wear-resistance performance, the depth of friction (8.192 μm) was only 14.4% of that in the substrate (56.918 μm), and the wear volume in the substrate (1.092 mm^3^) was 19.3 times larger than that in the WL-F15 coating (0.0567 mm^3^).

Figure 11 shows the SEM images and 3D morphologies of the wear surfaces of the TC4 substrate and coatings at a 20N load. In comparison to the wear surface at a 10 N load, the wear surface of the TC4 substrate (Figure 11(a,a1)) at a 20 N load exhibited deeper and longer furrows and a larger range of tear marks. Within the tear areas, larger spalling craters, deep furrows, and severe plastic deformation could be observed, indicating that adhesive and abrasive wear significantly increased the damage to the substrate. The 3D morphologies (Figure 11(a2)) showed deeper and longer furrows than the surface at a 10 N load. As shown in Figure 11(b,b1), the wear surface of the WL-F0 coating had deeper and longer furrows compared to the condition with a 10 N load. The number of scar regions on the wear surface of WL-F5 coating increased, as shown in Figure 11(c,c1). In addition, patches appeared on the wear surface, proving that the solid lubricant film with wear-reducing and wear-resistance properties exceeded its shear strength limit and broke down when the load was increased [40]. Due to the destruction of the solid lubricating film, it reduced the resistance to adhesive wear, resulting in more adhesive wear of the coating and an expansion of the scar regions. The wear mechanisms of the coatings are still abrasive and adhesive wear.

The wear surface of the WL-F10 coating (Figure 11(d,d1)) showed massive wear debris, furrows, micro-plowings, and cracks. The wear surface of the WL-F15 coating (Figure 11(e,e1)) lacked obvious furrows but exhibited widespread fish-scale-like patches, micro-plowings, and cracks. The oxidation film could not withstand the pressure and shear stress of the high load (20 N), leading to its damage and transformation into wear debris. The wear surface of the WL-F15 coating could not cause abrasive wear due to its extremely high surface hardness, such that the wear debris could only cause micro-plowings with a width less than 1 μm. The wear mechanisms of the WL-F10 coating are abrasive and adhesive wear; the wear mechanism of the WL-F15 coating is adhesive wear.

Figure 12a shows the wear track section curves of coatings and substrate at a load of 20 N, and Figure 12b shows the wear volume and depth of friction of coatings and substrates at a load of 20 N. The depth of friction, width of friction, and wear volume of the four groups of coatings slightly decreased compared with the substrate. The WL-F15 coating had the lowest depth of friction (61.479 μm), which is only 80% of that of the substrate (76.807 μm), and the WL-F10 had the lowest wear volume (1.041 mm^3^), which is only 61.2% of that of the substrate (1.701 mm^3^).

It can be seen that the wear resistance of the coatings was significantly improved at 10 N, but as the load increased to 20 N, the increment in wear resistance diminished. The WL-F10 and WL-F15 coatings exhibited a difference of an order of magnitude in wear volume under a load of 10 N apart. The transformation of their wear mechanism and changes in wear resistance are consistent with the wear-transition phenomenon in ceramic wear theory.

## 4. Discussion

### 4.1. Mechanism of Phase Formation

HSC Chemistry 9 was used to estimate the thermodynamic parameters of Ti_3_SiC_2_ and TiFeSi_2_ at 1000–3000 K and calculate the Gibbs free energy of the relevant processes at temperatures ranging from 1000 to 3000 K, as shown in Figure 13a,b. And the schematic diagram of the section microstructure before and after the phase transition process is shown in Figure 13c.

The possible reactions (2)–(12) described by the following equations could have occurred during the laser cladding process.
(2)SiC=Si+C
(3)Ti+C=TiC
(4)$$3Ti+Si+2C={Ti}_{3}SiC_{2}$$
(5)$$4Ti+6C+{Ti}_{5}{Si}_{3}=3{Ti}_{3}SiC_{2}$$
(6)$$5Ti+3Si={Ti}_{5}{Si}_{3}$$
(7)$$8Ti+3SiC=3TiC+5{Ti}_{5}{Si}_{3}$$
(8)$${Ti}_{3}SiC_{2}+Fe+SiC+Ti=TiFe{Si}_{2}+3TiC$$
(9)$${Ti}_{3}SiC_{2}+{Ti}_{5}{Si}_{3}+2Fe=TiFe{Si}_{2}+2TiC+4Ti$$
(10)$$3{Ti}_{5}{Si}_{3}+8Fe+7SiC=8TiFe{Si}_{2}+7TiC$$
(11)$$2Si+Ti+Fe=TiFe{Si}_{2}$$
(12)Si+C=SiC

The Gibbs free energy of Equation (2) is positive, indicating that the reaction was unable to occur spontaneously. Therefore, SiC participated in the reaction as a solid in the melt pool. The Gibbs free energies shown in Equations (3), (6) and (7) are all negative, indicating that all three reactions could occur spontaneously in this temperature range. Equation (7) provides the lowest Gibbs free energy in Figure 13a, which indicates that the formation of TiC and Ti_5_Si_3_ phases through the Ti with SiC was the most readily occurring reaction. Additionally, reactions (3) and (6) resulted in the formation of C and Si, which subsequently migrated within the melt pool. Reactions (4) and (5) signified the formation of the Ti_3_SiC_2_ phase. Equations (4) and (5) have a cross point at around 2100 K, which indicates that reaction (4) predominates at 1000–2100 K, while reaction (5) becomes dominant at 2100–3000 K. Reactions (8)–(11) represent the formation of the TiFeSi_2_ phase. The Gibbs free energies of Equations (8) and (9) are much lower than those of Equations (10) and (11), indicating that the conversion of Ti_3_SiC_2_ and Ti_3_SiC_2_ + Ti_5_Si_3_ into TiFeSi_2_ + TiC was more likely to occur. The Gibbs free energy of Equation (12) was negative, indicating that SiC could be formed via this reaction.

By combining XRD, EDS results, and HSC chemical reaction calculations, the phase transition process during the preparation of the working layer can be inferred, as shown in Figure 13c. The process of preparing Ti-xFe-80SiC (vol.%) (x = 0, 5) coatings on the transition layer caused some of the coarser dendrite Ti_5_Si_3_ in the top region of the transition layer to transform into Ti_3_SiC_2_. During the process of preparing Ti-xFe-80SiC (vol.%) (x = 10, 15) coatings on the transition layer, the original Ti_5_Si_3_ and Ti_3_SiC_2_ began to transform toward TiFeSi_2_ and TiC, and a small amount of SiC was generated at the TiC grain boundary. These alterations in phase structure and microstructure resulted in notable performance enhancements, including a substantial increase in wear resistance and a significant improvement in hardness.

The phase-transition phenomenon indicated that during the preparation process of the working layer, only the top region of the transition layer had transformed into a melt pool. It can be explained that the high melting point of the ceramic phase makes it difficult to completely melt in the melting process, thereby inhibiting the increase in the melting depth of the working layer. If the working layer is directly prepared on the substrate, a large number of Si and C elements will diffuse into the substrate, resulting in a high dilution of the cladding layer, which will seriously affect the performance of the material. Thus, the transition layer plays an important role in reducing the dilution rate of the matrix in the coating with high-volume-fraction ceramics prepared via laser cladding.

### 4.2. Wear Mechanism and Wear Transition

In frictional wear experiments of the coatings and substrate, distinct wear mechanisms govern the processes of wear damage. This study primarily addresses three main wear mechanisms: abrasive wear, adhesive wear, and oxidation wear. The schematic diagram of these wear mechanisms is shown in Figure 14. Abrasive wear is a fundamental mechanism in material wear processes, involving the tangential cutting of the material surface using a hard friction pair and generating wear debris during friction, which results in the formation of abrasive scratches [41]. As shown in Figure 14a, indications of abrasive wear include furrows and the accumulation of plastic deformation on both sides. Adhesive wear is another one of the primary mechanisms in material wear processes. This mechanism occurs when material particles partially detach due to the adhesion and separation between the friction pair and the material contact surfaces during wear [42]. As shown in Figure 14b, indications of adhesive wear include spalling craters and scar regions. Oxidation wear is a form of chemical wear where oxidation occurs as the constituents of the material surface chemically react with atmospheric oxygen [43]. It serves as a classic example of wear due to chemical reactions, often leading to the formation of an oxidation film (Figure 14c), which serves as clear evidence of this wear process.

The WL-F10 and WL-F15 coatings exhibited significantly different wear mechanisms and substantially varied wear resistance at loads of 10 N and 20 N. Considering the high-volume-fraction of ceramics within the coatings, this phenomenon could be explained by applying the concept of wear transition from ceramic wear theory. The frictional wear behavior of structural ceramics can be categorized into several distinct stages based on the severity of wear, including mild, medium, and severe stages [44]. The change in the wear mechanism between each stage is called wear transition. Multiple parameters, such as load, speed, sliding distance, sliding time, and temperature are closely linked to wear volume [45,46,47,48]. The primary reason for the transition from mild wear to severe wear in WL-F10 and WL-F5 coatings was the increase in the load. The higher radial forces at elevated loads resulted in the formation of cracks in the oxidation film. As the wear time increases, these cracks gradually extended deeper and outward, and the fracture caused by the cracks led to the removal of the material under high contact stress. This led to the appearance of wear morphology such as oxidation film rupture, intergranular fracture of brittle ceramic phases, scar regions, and fish-scale-like patches. The surfaces of WL-F10 and WL-F15 coatings underwent a cyclic wear process of cracking–peeling–cracking under a high load, during which the coating material had been extensively removed, resulting in a significant decrease in wear resistance.

## 5. Conclusions

The Ti-xFe-SiC in situ composite ceramic coatings were prepared on the TC4 substrate using laser cladding. The microstructure and properties of the laser cladding coating were investigated, and the following conclusions were obtained.

The transition layer exhibits good metallurgical bonding with the substrate. It can effectively prevent the direct contact between the working layer and the substrate from causing an excessive dilution rate and ensure that there is a high content of ceramic phase in the working layer.With increasing content of Fe, the hardness of the working layer also increases obviously. The WL-F15 coating, exhibiting the most substantial increase in hardness, achieves a maximum hardness of 1842.9 HV_1_, which is five times higher than that of the substrate.Compared to the substrate, all the coatings exhibited a lower coefficient of friction (CoF), depth of friction, and wear volume. It was demonstrated that the coatings have a certain degree of wear-reducing and wear-resistance performances. The WL-F15 coating has the best wear resistance under a 10 N load. Under a load of 10 N, the wear volume and depth of friction of the WL-F15 coating was only 5.2% and 14.4% of that in the substrate. Under a load of 20 N, the WL-F10 coating has the lowest wear volume, and the WL-F15 coating has the lowest depth of friction. Overall, WL-F10 has the best wear resistance under a 20 N load. The experiments showed that the coatings have excellent wear-resistance properties at low loads, while the large gap in wear resistance between high and low loads can be attributed to the wear transition.

## Figures and Tables

**Figure 1 materials-17-00100-f001:**
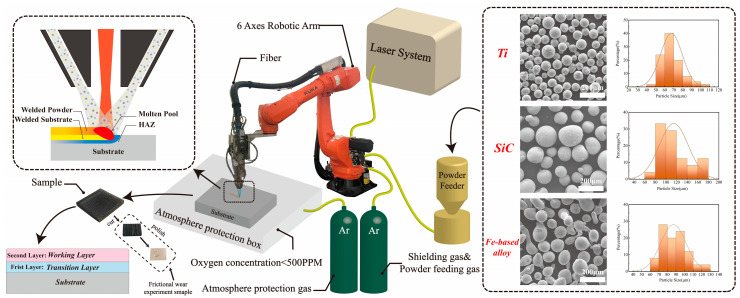
The schematic diagram of the laser cladding process.

**Figure 2 materials-17-00100-f002:**
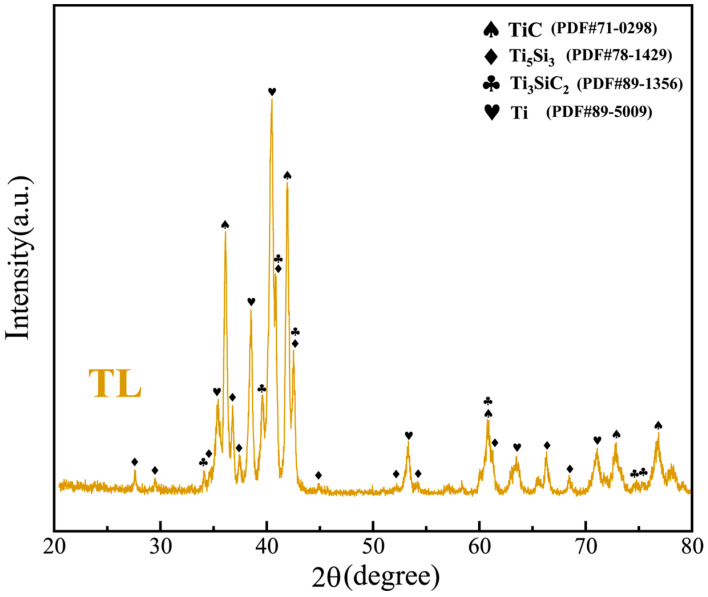
XRD results of Ti-40SiC (TL) coating.

**Figure 3 materials-17-00100-f003:**
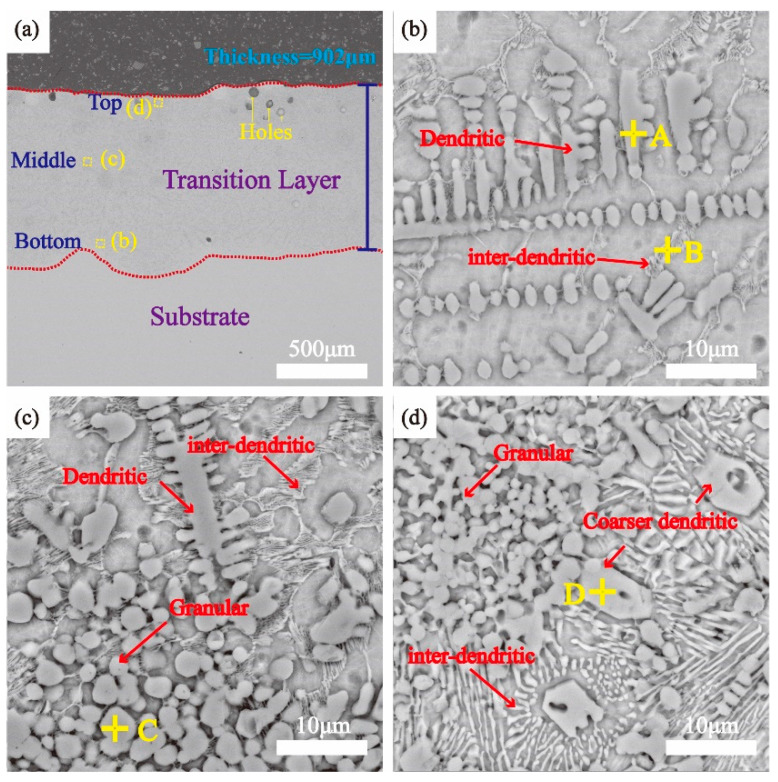
SEM images of the cross-sectional microstructure of the coating: (**a**) TL coating; (**b**–**d**) are enlargements of the areas shown in panels.

**Figure 4 materials-17-00100-f004:**
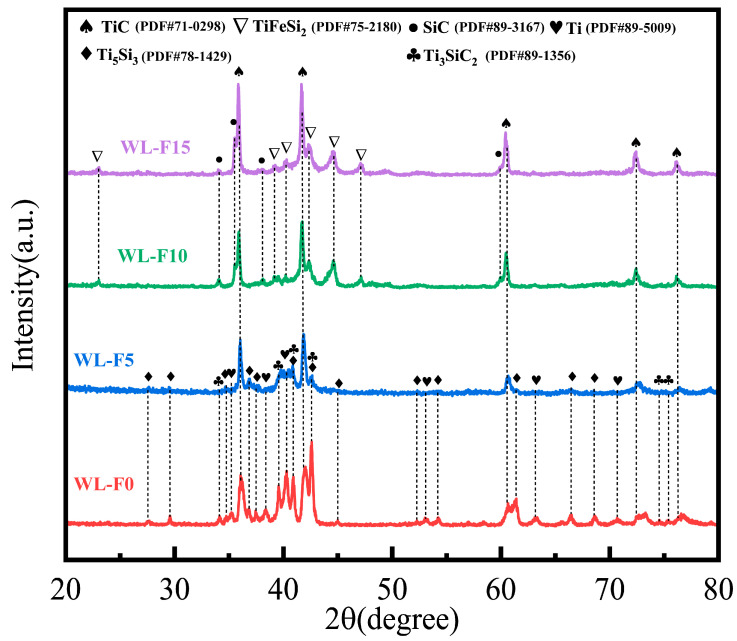
X-ray diffraction results of the Ti-80SiC(WL-F0), Ti-5Fe-80SiC(WL-F5), Ti-10Fe-80SiC(WL-F10), and Ti-15Fe-80SiC(WL-F15) coatings.

**Figure 5 materials-17-00100-f005:**
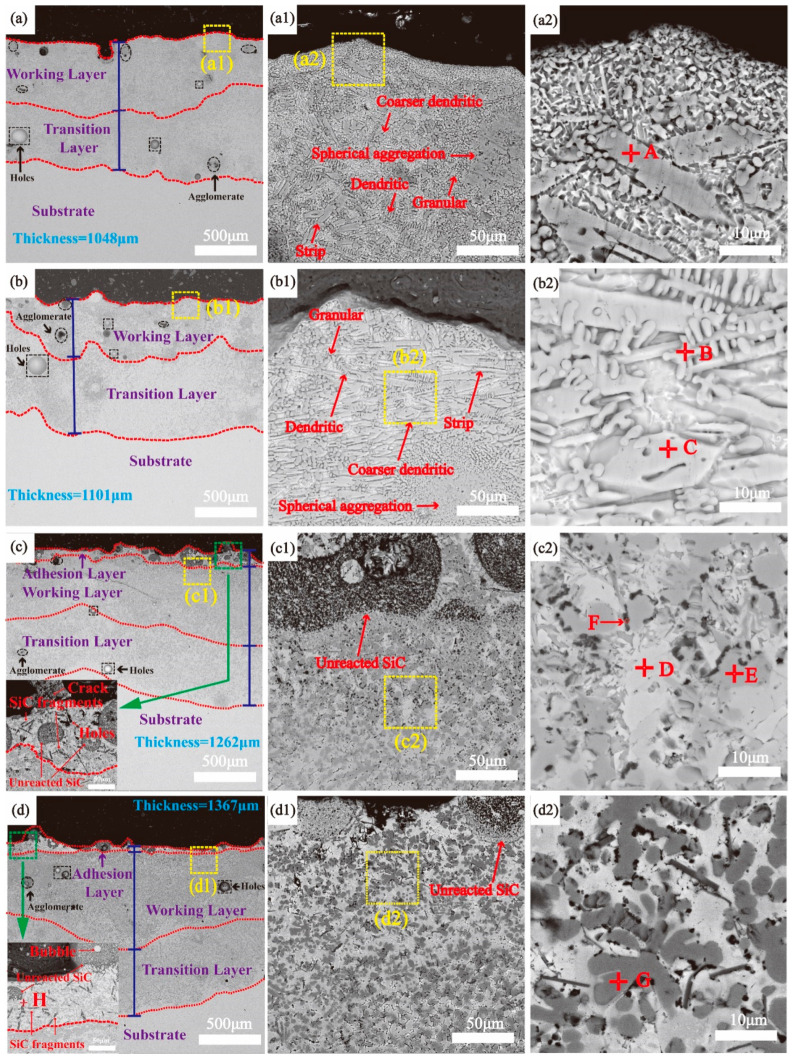
SEM images of the cross-sectional microstructure of the coating: (**a**) WL-F0, (**b**) WL-F5, (**c**) WL-F10, (**d**) WL-F15, (**a1**–**d1**) are enlargements of the areas shown in panels (**a**–**d**), respectively; (**a2**–**d2**) are enlargements of the areas shown in panels (**a1**–**d1**), respectively.

**Figure 6 materials-17-00100-f006:**
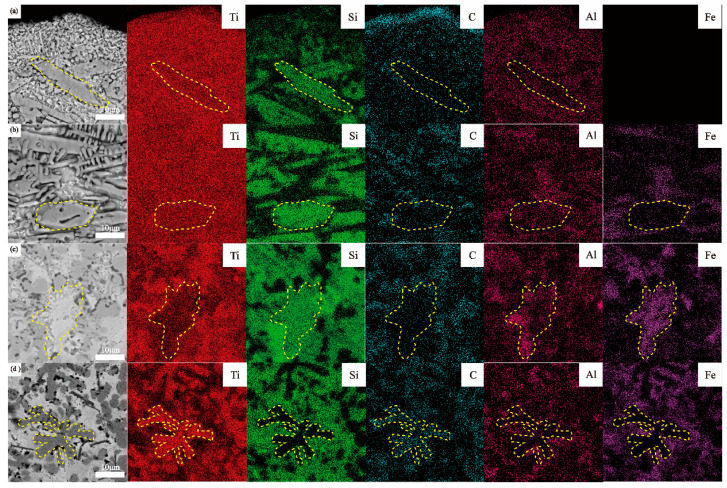
EDS map scanning of the element distribution of four coatings: (**a**) WL-F0, (**b**) WL-F5, (**c**) WL-F10, and (**d**) WL-F15.

**Figure 7 materials-17-00100-f007:**
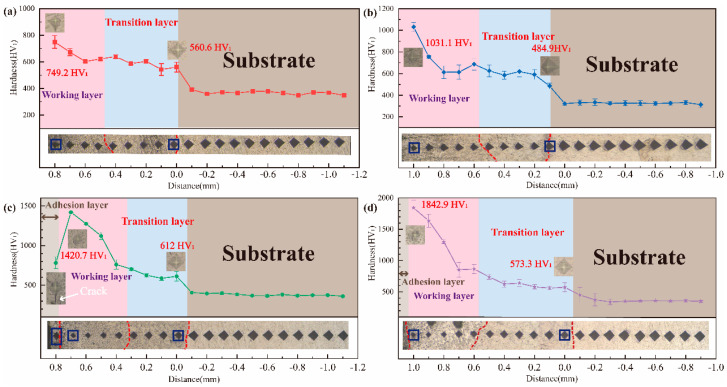
Vickers hardness of the cross–sectional coatings: (**a**) WL-F0, (**b**) WL-F5, (**c**) WL-F10, and (**d**) WL-F15.

**Figure 8 materials-17-00100-f008:**
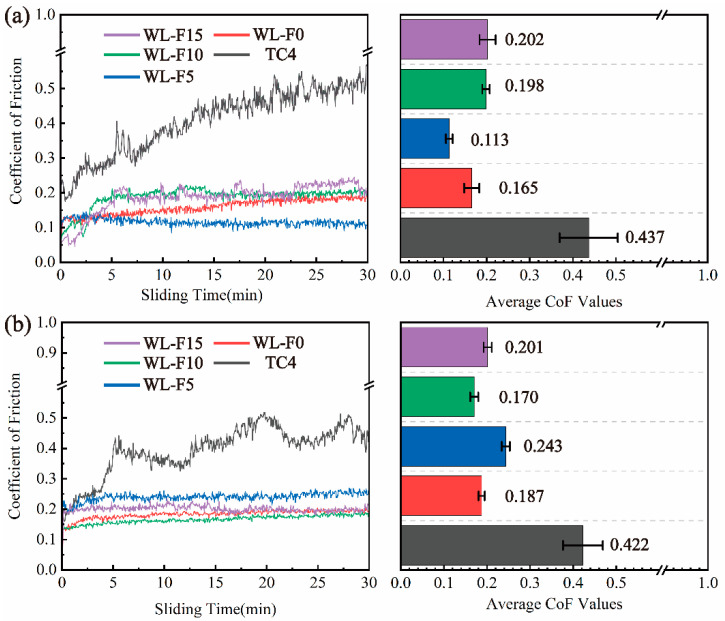
The coefficients of friction (CoF) curves and average CoF values of the substrate and coatings: (**a**) at a 10N load and (**b**) at a 20 N load.

**Figure 9 materials-17-00100-f009:**
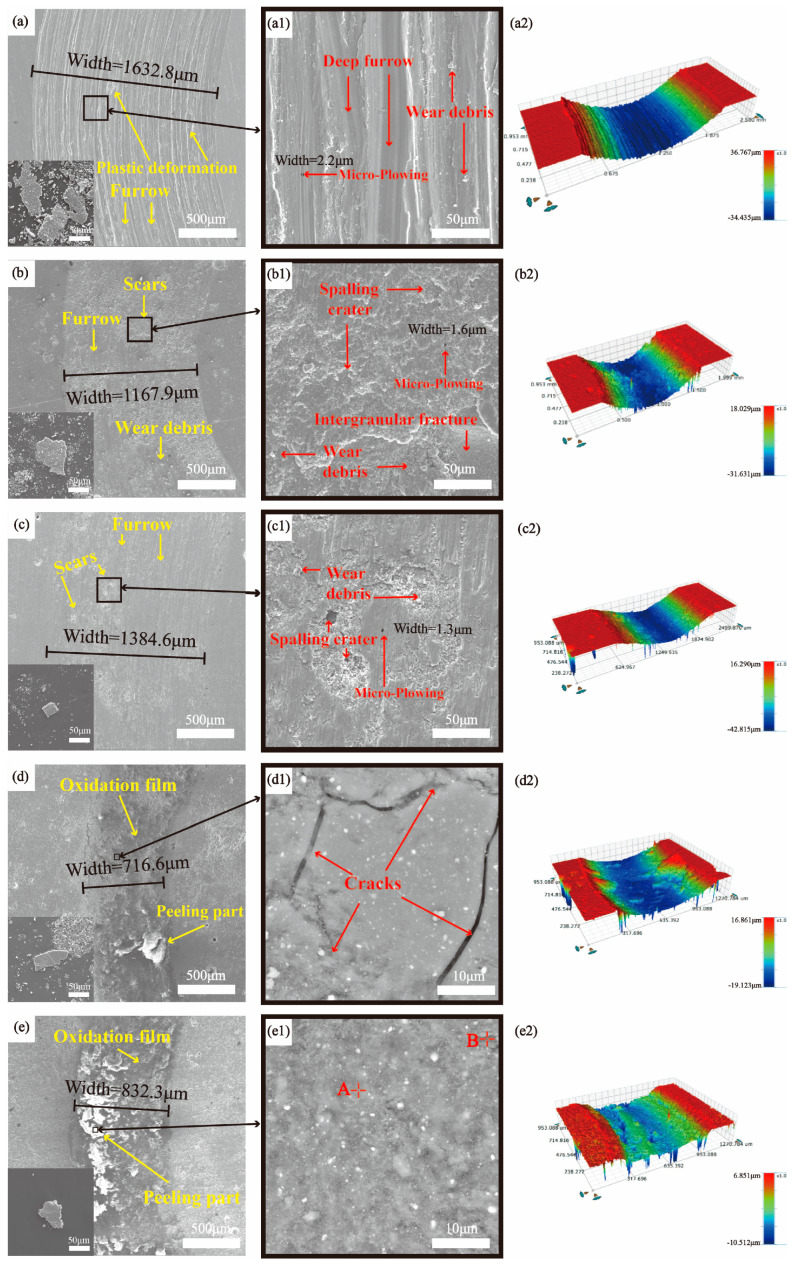
SEM images of the wear surfaces on TC4 substrate and coatings at 10 N load: (**a**) TC4, (**b**) WL-F0, (**c**) WL-F5, (**d**) WL-F10 and (**e**) WL-F15; (**a1**–**e1**) are enlargements of the areas shown in panels (**a**–**e**), respectively; 3D morphologies of (**a2**) TC4, (**b2**) WL-F0, (**c2**) WL-F5, (**d2**) WL-F10, and (**e2**) WL-F15.

**Figure 10 materials-17-00100-f010:**
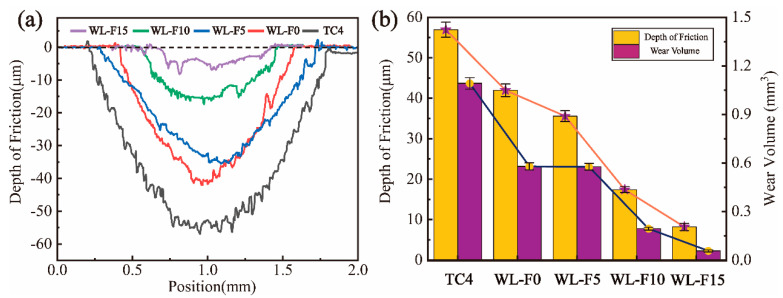
(**a**) The wear track section curves of the coatings and TC4 substrate at a 10 N load; (**b**) the depth of friction and wear volume of the coatings and TC4 substrate at a 10 N load.

**Figure 11 materials-17-00100-f011:**
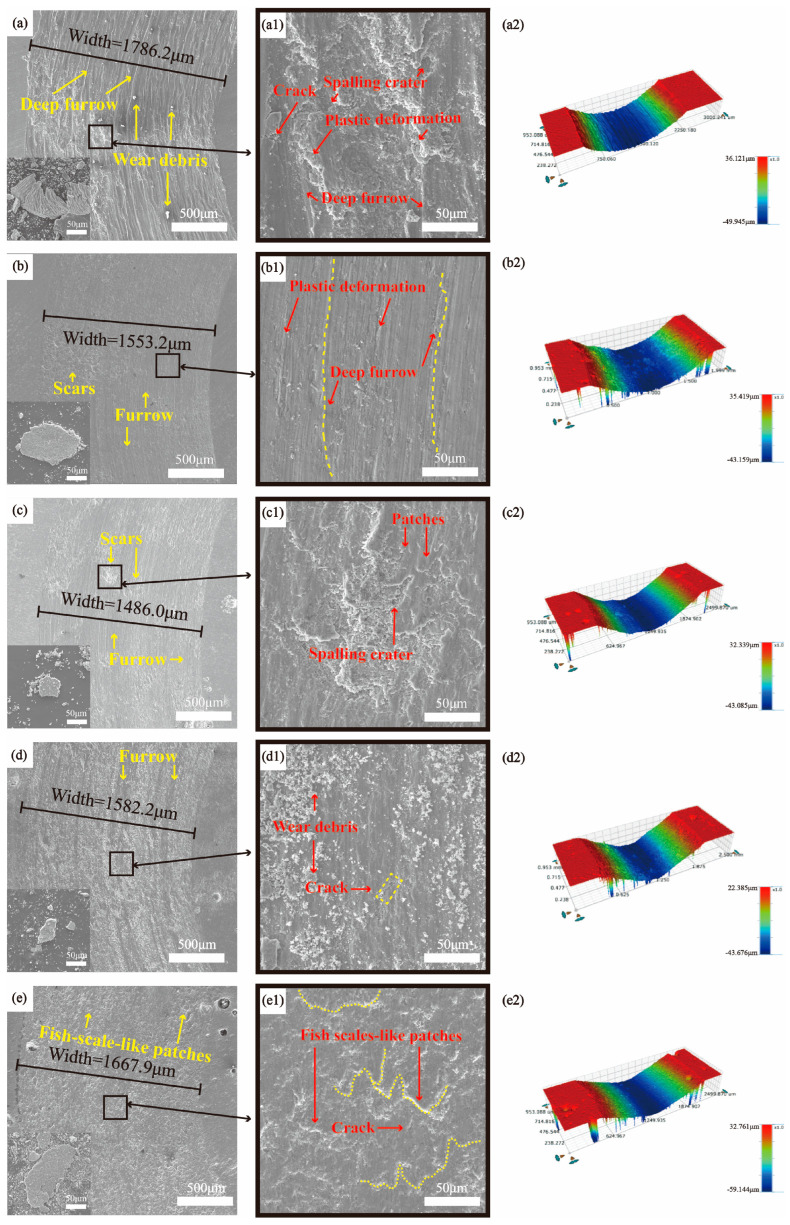
SEM images of the wear surfaces on TC4 substrate and coatings at 20N load: (**a**) TC4, (**b**) WL-F0, (**c**) WL-F5, (**d**) WL-F10 and (**e**) WL-F15; (**a1**–**e1**) are enlargements of the areas shown in panels (**a**–**e**), respectively; 3D morphologies of (**a2**) TC4, (**b2**) WL-F0, (**c2**) WL-F5, (**d2**) WL-F10, (**e2**) WL-F15.

**Figure 12 materials-17-00100-f012:**
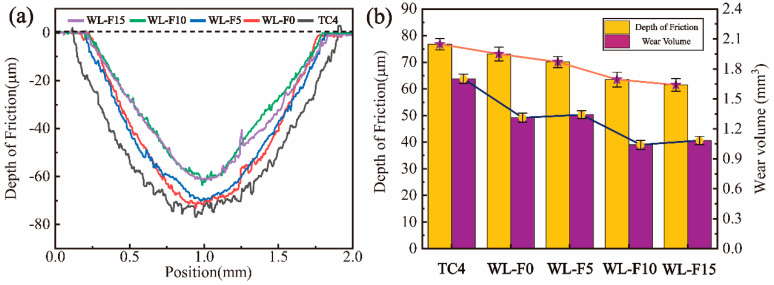
(**a**) The wear track section curves of coatings and substrate at a 20 N load; (**b**) the depth of friction and wear volume of the coatings and TC4 substrate at a 20 N load.

**Figure 13 materials-17-00100-f013:**
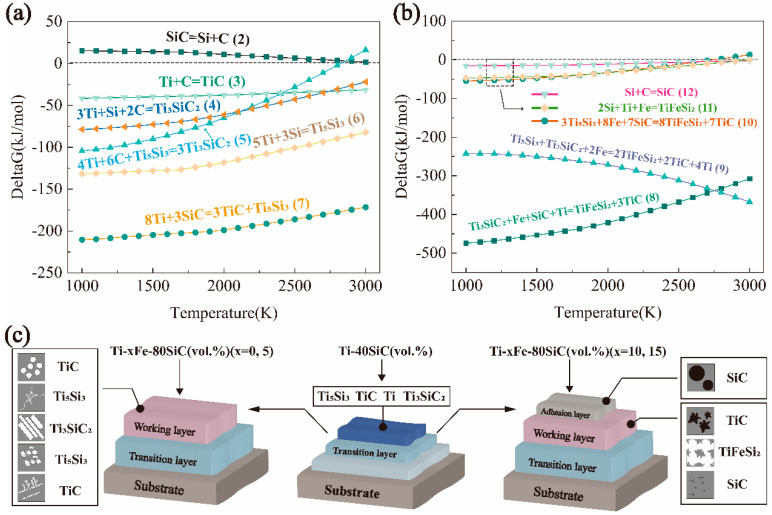
(**a**) Gibbs free energy curves of Ti–Si–C system reactions; (**b**) Gibbs free energy curves of Ti–Fe–Si–C system reactions; (**c**) the schematic diagram of section microstructure before and after the phase transition process.

**Figure 14 materials-17-00100-f014:**
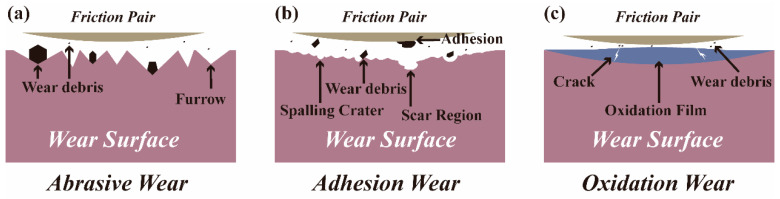
Schematic diagram of wear mechanisms: (**a**) abrasive wear, (**b**) adhesion wear, and (**c**) oxidation wear.

**Table 1 materials-17-00100-t001:** Powder composition for Ti-Fe-SiC (vol.%) composite coatings.

Coatings	Compositions	Content (vol.%)		
		SiC	Ti	Fe-Based Alloy
TL	Ti-40SiC	40	60	0
WL-F0	Ti-80SiC	80	20	0
WL-F5	Ti-5Fe-80SiC	80	15	5
WL-F10	Ti-10Fe-80SiC	80	10	10
WL-F15	Ti-15Fe-80SiC	80	5	15

**Table 2 materials-17-00100-t002:** Point composition analysis of the coating tissue of Figure 3 (at. %).

Point	Element					Possible Phase
	Ti	Si	C	Al	Others	
A	49.57	1.05	48	0.84	0.54	TiC
B	67.65	11.53	13.49	5.17	2.16	Ti_5_Si_3_
C	49.73	0.18	48.92	0.14	1.03	TiC
D	58.69	30.25	7.32	3.2	0.54	Ti_5_Si_3_

**Table 3 materials-17-00100-t003:** Point composition analysis of the coating tissue shown in Figure 5 (at. %).

Point	Elements						Possible Phase
	Ti	Si	C	Fe	Al	Others	
A	57.63	15.01	24.32	-	2.38	0.66	Ti_3_SiC_2_
B	47.22	0.8	50.99	0.21	0.37	0.41	TiC
C	58.58	31.53	6.36	0.23	1.93	1.37	Ti_5_Si_3_
D	18.6	35.4	26.75	14.1	1.16	3.99	TiFeSi_2_
E	38.83	2.09	57.48	0.15	0.31	1.14	TiC
F	12.94	25.36	53.47	2.22	4.9	1.11	SiC
G	42.4	0.34	55.55	0.18	0.09	1.44	TiC
H	1.78	43.4	53.86	0.24	0.44	0.28	SiC

**Table 4 materials-17-00100-t004:** Point composition analysis of the coating tissue of Figure 9 (at. %).

Point	Element						Possible Phase
	Ti	Si	O	C	Al	Others	
A	10.37	20.1	55.51	10.31	1.62	2.09	SiO_2_ + TiO_2_
B	12.33	19.9	48.26	16	1.55	1.96	SiO_2_ + TiO_2_

## Data Availability

The data presented in this study are available upon reasonable request from the corresponding author.

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
