# Peer review of "Microstructure and Wear Behavior of Ti-xFe-SiC In Situ Composite Ceramic Coatings on TC4 Substrate from Laser Cladding"

_materials, 2023, doi:10.3390/ma17010100_

Round 1

Reviewer 1 Report

Comments and Suggestions for Authors

It is suggested to deeper, into the discussion section, explaining how the transition layer participates in reducing the dilution rate of the matrix in the coating.

Is there an optimal amount of Fe to add to produce a defect-free material?

Reviewer 2 Report

Comments and Suggestions for Authors

Research about Microstructre and wear behavior of Ti-xFe-SiC in-situ composite ceramic coatings on TC4 substrate by laser cladding has been done by X.J. Zhao et al. The microstructure of the 5 specimens with different Fe addition levels has been comprehensively discussed as well as the hardness results. However, there are some important issues that have not been well explained and even seem to be wrong when reading the wear volumes obtained. Therefore, this should be investigated and clarified again. Here are some issues that can be used to improve the validity of this study

1.       In abstract section, it is important to state the urgency of this present study. In addition, it is suggested to describe where this material can be applied in the abstract section.

2.       Please carefully check the space of each sentence in the introduction.

3.       What is the XRD data based that used in this study?

4.       The condition of materials surface (friction/roughness) before the test can influence the wear performance. Did Authors measure it? Or Authors just assumed that all material surface roughness is same? Please provide it. Besides that, please provide the picture of material surface after cladding from the above view to show the quality of the laser cladding of this study.

5.       The reference of wear volume formula (page 13) should be provided.

6.       Why the wear volume of WLF0 and WL-F5 with 10N and 20 N is relatively same while the depth friction, hardness of both materials is different?

7.       Why the wear volume of WL-F15 is lower than WL-F10 with 10N but it is relatively same 20N?

8.       Figure 12 shows that the wear volume of WL-F10 and WL-F15 is relatively same or slightly increase from WL-F10 to WL-F15. Therefore, the conclusion of number 2 that state the WL-F15 has the best wear resistance is wrong or not accurate. Please give clarify this finding with reasonable reason and evidence.

Comments on the Quality of English Language

Minor editing of English language required

Reviewer 3 Report

Comments and Suggestions for Authors

1.     Define TC4 when it was used first in the abstract as well as in the main text.

2.     Remove empty line in the caption of figures 6,7,8,9,11 and 14.

Round 2

Reviewer 2 Report

Comments and Suggestions for Authors

Thank you for your efforts to improve the quality of your article. The revised version is acceptable for publication.